# Successful Integration of HIV PrEP in Primary Care and Women’s Health Clinical Practice: A Model for Implementation

**DOI:** 10.3390/v15061365

**Published:** 2023-06-13

**Authors:** Eunice Casey, Emma Kaplan-Lewis, Kruti Gala, Rebecca Lakew

**Affiliations:** 1HIV Services, Office of Ambulatory Care and Population Health, NYC Health and Hospitals, 50 Water Street, 6th Floor, New York, NY 10004, USA; eunice.casey@nychhc.org (E.C.); galak@nychhc.org (K.G.); 2Chronic Diseases and Prevention, Office of Ambulatory Care and Population Health, NYC Health and Hospitals, 50 Water Street, 6th Floor, New York, NY 10004, USA; lakewr@nychhc.org

**Keywords:** HIV prevention, PrEP care model, PrEP implementation, PrEP for women, PrEP expansion

## Abstract

Ending the HIV Epidemic is contingent upon the increased utilization of pre-exposure prophylaxis (PrEP). The majority of PrEP in the United States is prescribed in specialty care settings; however, to achieve national implementation goals, it is necessary to expand PrEP services in primary care and women’s health clinics. To this end, a prospective cohort study was conducted of health care providers participating in one of three rounds of a virtual program aimed at increasing the number of PrEP prescribers in primary care and women’s health clinics within the NYC Health and Hospitals network, the public healthcare system of New York City. Provider prescribing behavior was compared at pre-intervention (August 2018–September 2019) and post-intervention (October 2019–February 2021). Among 104 providers, the number prescribing PrEP increased from 12 (11.5%) to 51 (49%) and the number of individual patients on PrEP increased from 19 to 128. The program utilized clinical integration models centering on existing STI management workflows and was associated with increased numbers of PrEP prescribers and volume of prescriptions in primary care and women’s health clinics. The dissemination of similar programs could support national scale-up of PrEP.

## 1. Introduction

In order to meet national [1] and New York State [2] goals for ending the HIV epidemic (ETE), the prevention of incident HIV infections is key. While pre-exposure prophylaxis (PrEP) is a highly effective intervention for decreasing HIV transmission, the most recent available data show that only 25% of eligible patients in the U.S. received PrEP in 2020, an increase from 18% in 2018 [3,4]. There are significant disparities in who is aware of and receives PrEP, with white men who have sex with men residing in the Northeastern U.S. most likely to access PrEP [4,5,6,7], and inequities in PrEP access by race have worsened over time [3,8]. Similar to many healthcare systems nationwide, at NYC Health and Hospitals (NYC H+H), the majority of PrEP is prescribed in specialty care settings, while sexually transmitted infections (STI) are often treated in the primary care setting [9,10,11]. It should be noted that ETE goals for PrEP uptake far exceed the capacity of specialty care clinics [11,12,13] and many patients do not want to be referred out of primary care just for PrEP services. Specialty care settings, such as a sexual health clinics, LGBT care clinics or infectious disease/HIV clinics [14,15,16,17], are valuable service settings for patients who prefer these, but some individuals feel stigmatized [18] and inconvenienced when seeking sexual health care in a specialty location. Therefore, this program sought to provide a model for integrating PrEP services alongside existing STI screening and treatment in primary care and women’s health clinics, increasing the possibility of reaching ETE goals for PrEP use.

NYC H+H includes a network of 11 acute care hospitals and more than 30 federally qualified health centers that make up the safety net public healthcare system in New York City. NYC H+H provides care to over 14,000 New Yorkers living with HIV (38% female sex) as well as over 19,000 New Yorkers (32% female sex) who could benefit from targeted HIV prevention efforts due to the increased risk of HIV acquisition. In 2019, NYC H+H Office of HIV Services received grant funding from the New York State Department of Health, AIDS Institute, to evaluate an online provider education and support program. The program was aimed at increasing the number of primary care and women’s health providers offering PrEP as well as expanding overall access to PrEP services throughout the primary care and women’s health clinics of the NYC H+H network. The present study attempts to answer whether an online provider education curriculum in conjunction with a series of customized support tools would be effective at increasing the number of PrEP prescribers and patients being prescribed PrEP.

## 2. Materials and Methods

The intervention consisted of three support tools tested in discrete combinations across three distinct implementation rounds over the study period, as detailed in Figure 1. The proposed structure included a tiered system for PrEP care, modeled after routine STI care, where uncomplicated PrEP care would be provided in primary care/women’s health clinics and complex care/cases would be supported by specialty providers either within those clinics or through referral. The baseline intervention consisted of a six-part PrEP curriculum designed to follow the Project ECHO (Extension for Community Healthcare Outcomes) model. Three implementation rounds of the curriculum were administered with new enrollees added at each round. Once enrolled, providers were allowed to participate in more than one round of the curriculum. The second support was specific lists for providers of PrEP-eligible patients (individuals diagnosed by the provider over the preceding six months with an STI). These lists were given to providers enrolled in the first round of the curriculum. Lastly, one hospital tested the value of a dedicated staffer embedded first within their women’s health and subsequently within their primary care clinics to support a customized PrEP implementation plan based on clinic-specific resources and patient needs. The dedicated staffer was introduced at the beginning of the second round of the curriculum and continued working until the beginning of the COVID-19 pandemic in New York City, which occurred between sessions four and five of Round 2. Notably, COVID-19-related shutdowns resulted in a five-month pause in the program.

We examined outcomes based on which intervention supports were utilized by providers (completed ECHO: defined as participation in three or more sessions; curriculum only: defined as participation in two or less sessions; PrEP-eligible patient lists provided; and implementation staff support).

Patients and the public were not involved in the design or conduct of this research because the intervention was focused on clinical providers. Both patients (through involvement in statewide HIV consumer advisory boards) and the New York State HIV provider community were involved in the dissemination of these research results.

### 2.1. Curriculum Detail

The PrEP ECHO curriculum focused on the patient services most needed for a tiered level of care between primary care/women’s health settings and specialty providers. The curriculum was developed with input from clinical leads within NYC H+H as well as the NYC Department of Health and Mental Hygiene (DOHMH) Bureau of Hepatitis, HIV, and Sexual Health. The ECHO model is structured to facilitate case discussion and provide specialist input to support integrating clinical interventions in primary care clinics [19]. Six bi-weekly one-hour PrEP ECHO online sessions were structured to include 20 minutes each of a didactic presentation, case presentation, and discussion period. Three rounds of the six-session curriculum were completed as part of this evaluation, with Round 1 launching in September 2019 and Round 3 finishing in October 2020. The curriculum was divided into the following sessions: “Initial Medical Assessments for PrEP”, “Following Patients on PrEP”, “Special Topics Related to PrEP for Women”, and “Sexual History—the GOALS Framework” [20]. This framework is designed to elicit the most useful sexual behavior information from a patient in a non-stigmatizing and sex positive way, with the explicit purpose of informing clinical care. The last two sessions focused on the integration and implementation of PrEP services into primary care/women’s health clinics, with topics dedicated to the following: “Clinical Workflows—Who Does What and When” and “Implementing PrEP—Bringing It All Together”. Copies of session materials were sent to all enrolled individuals regardless of attendance. All sessions occurred on the Webex online virtual conference platform. Provider recruitment targeted adult primary care and women’s health clinics across NYC H+H and was promoted by local facility clinical leadership. “Primary Care” clinics include adult medicine clinics where internal medicine and family practice providers see patients. “Women’s Health” clinics include adult obstetrics and gynecology clinics. Participation was voluntary.

### 2.2. Intervention

There were three rounds of the PrEP ECHO curriculum. Providers enrolled in Round 1, whose panels could be matched to existing electronic medical record (EMR) STI lab data, had access to an EMR decision support tool identifying patients on their panel with one or more STI diagnoses within the previous six months. These patients were classified as “PrEP-Eligible” and lists of their names were provided just prior to session five of the curriculum, where follow-up processes were reviewed. A dedicated implementation support staffer was hired to work in a single hospital location and support PrEP integration first within women’s health and then within primary care. It should be noted that in New York State, PrEP medication is available at minimal or no cost to the patient, whether through insurance or pharmaceutical prescription assistance programs, and as such, any patient prescribed PrEP was able to access medication. Navigation through different payment options for PrEP medication was a central component of the dedicated implementation support staffer’s work to build a customized plan for the clinic.

Due to the three different supports provided in this study, four separate groups were analyzed: (1) completed PrEP ECHO program (defined as attending three or more sessions); (2) curriculum access only (defined as attending two or fewer sessions); (3) data support in the form of PrEP eligible patient lists; and (4) on-site implementation staffer support. All participants fell into either group one or group two. Participants that had the additional support of PrEP-eligible patient lists or a dedicated implementation support staffer were reviewed separately to examine the impact of these additional supports.

### 2.3. Analysis

Analysis focused exclusively on primary care and women’s health providers who maintained a panel of continuity patients at their respective clinical sites. All other participating clinicians and staff, including residents, were excluded from data analysis. We evaluated the outcomes related to change in number of PrEP prescribers, the total number of prescriptions, change in prescription volume by prescriber, and the number of patients on PrEP during a pre-intervention timeframe from August 2018 through September 2019 (14 months) as compared to a post-intervention timeframe from October 2019 through February 2021 (17 months). In addition to calculating means and proportions, bivariate analysis was performed using McNemar’s test to evaluate for a change in the number of PrEP providers pre- versus post-intervention after converting PrEP prescription to a binary response of yes or no. Wilcoxon’s signed-rank test was used for evaluating the change in the number of prescriptions pre- and post-intervention since the data are paired and continuous and not normally distributed. As a secondary analysis, we evaluated these outcomes by level of provider participation in the intervention curriculum and supports utilized (i.e., completed PrEP ECHO, curriculum access only, PrEP-eligible patient lists, and implementation staffer support). Missing data for PrEP prescriptions were assumed to equal zero for both pre-intervention and post-intervention numbers.

## 3. Results

There were 190 participants who enrolled in the PrEP ECHO, of whom 104 were included in the analysis. Eighty-six were excluded because they were either not primary care or women’s health providers or because they were medical residents. Of the 104 participants analyzed, 80 were primary care (77%) and 24 were women’s health providers (23%). At pre-intervention, there were 32 PrEP prescriptions among participating providers, which increased to 297 post-intervention (Table 1). The average number of prescriptions per provider increased from 0.3 at pre-intervention to 2.85 post-intervention. The number of individual patients receiving PrEP increased from 19 to 128 post-intervention. Table 2 shows the increase in women’s health providers prescribing PrEP (2 to 10, *p* = 0.013), as well as the increase in the number of individual PrEP prescriptions written by women’s health providers (3 to 43, *p* = 0.004). The magnitude of increase was similarly large for primary care providers with an increase from 10 to 41 providers prescribing PrEP (*p* < 0.001) and an increase from 29 to 247 individual PrEP prescriptions (*p* = 0.009).

The effects of the intervention on the number of participating providers prescribing PrEP are shown in Table 3, with the number of women’s health providers prescribing PrEP increasing from 2 (8%) to 10 (42%) and the number of primary care providers prescribing PrEP increasing from 10 (13%) to 41 (51%). The number of providers prescribing PrEP who completed the ECHO curriculum (attending three or more sessions) increased from 4 (11%) to 18 (51%), which was similar to the increase seen among providers prescribing PrEP who attended two or fewer sessions (labeled Curriculum Only), with the number of PrEP prescribers increasing from 8 (12%) to 33 (48%). Providers who received PrEP-eligible patient lists in addition to the ECHO curriculum increased the number of PrEP prescribers from 6 (20%) to 15 (50%). Providers who worked at the clinic site where an implementation support staffer was embedded increased the number of PrEP prescribers from 0 to 8 (67%) after the intervention. While all components of the intervention (PrEP ECHO curriculum, PrEP-eligible lists, implementation support staffer) led to increases in the number of providers prescribing PrEP, the numbers of providers in each of these intervention groups were too small to carry out additional analyses.

## 4. Discussion

In this multimodal initiative to expand access to PrEP in primary care and women’s health settings, increases in PrEP provision were observed across all measures: expanding the number of PrEP prescribers, increasing prescription volume by prescriber, and increasing the number of patients on PrEP. Given the need to dramatically increase the volume of individuals accessing PrEP services to achieve ETE goals, novel approaches, including integrating PrEP in primary care clinical settings, are required to expand access beyond specialty clinics. The implementation of the model explored in this intervention would support using a variety of tools in combinations that are appropriate for the targeted clinics. The tools that were effective in this model included an online educational curriculum, the dissemination of curriculum materials beyond those attending educational sessions, the use of provider-specific PrEP-eligible patient lists and an implementation support staffer. Sexual health and HIV clinics continue to be a valued and necessary resource for PrEP programs, but they do not have the capacity to provide services for the large number of patients that could benefit from PrEP care [13,21,22].

Both specialty care and primary care/women’s health clinics have a role to play in expanding PrEP access. This is not a question of PrEP at one care site instead of the other but of maximizing the opportunities for each type of care location to meet the needs of different patients and support the scale-up of PrEP services. The studied program reached patients engaged in primary care/women’s health and integrated basic PrEP services as part of their routine care within the clinical setting the patient independently choose. Supplementing this effort is the presence of a network of sexual health/LGBT and/or HIV clinics co-located within the same facilities where expanded sexual health care is available for patients seeking interventions beyond basic STI and PrEP care. This tiered level of support is increasingly common in primary care settings where providers can expand access to required services while also supporting the identification and referral of individuals that need more specialized resources [23]. Additionally, specialty providers are a valued resource to primary care providers expanding their scope of practice. Bringing this tiered health structure to PrEP care is a natural progression of the existing care systems for STI diagnosis and treatment and provides a model for improving sexual health services and access.

Earlier efforts to expand PrEP services into primary care settings often sought to take the PrEP care model from specialty clinics and superimpose it onto primary care sites [24], not recognizing the service demands and workflows of primary care or women’s health practices. The approach taken by this study began with the concept of a relevant but low-threshold definition for PrEP eligibility, namely a recent STI diagnosis. Understanding that many patients already seek STI care within primary care/women’s health settings and that diagnosis of, or concern regarding, an STI are by themselves suggestive of sexual activity with an elevated risk of HIV acquisition. Utilizing the diagnosis of a recent STI as the starting point for discussing and offering PrEP enables PrEP to fit into sexual healthcare that is already occurring in primary care and women’s health clinics. In addition, not all individuals want to go to a specialty care setting for their sexual health and HIV prevention needs [8,13] and limiting PrEP services to these locations deters these individuals from accessing PrEP services, furthering the equity issue of who can access PrEP. There are numerous barriers to PrEP uptake that have been reported [25,26,27,28], such as stigma, cost concerns, and knowledge deficit. Referring out or transferring care away from primary care/women’s health clinics creates an additional barrier that contributes to decreased PrEP uptake.

Flexible care systems that address both patient preferences and the existing primary care/women’s health infrastructure may have more success with meeting clinical endpoints, such as PrEP utilization, and thus decreasing HIV incidence [3,29,30,31]. The intervention components chosen for this study were selected based on their demonstrated effectiveness across a range of primary care settings [23,32]. The success of the different intervention components in this project suggests that implementation support for PrEP may be most effective when including multiple options for education and assistance. Our analysis shows that all the interventions were associated with increases in PrEP use and the variety of interventions appears to have allowed different providers and clinic sites to scale their needs and learning styles to the most locally relevant approaches.

The major limitation in this project was the COVID-19 pandemic, which resulted in the NYC-wide closure of non-emergent in-person clinical care between the fourth and fifth sessions of Round 2 and a five-month pause in the program. Once the pause ended, provider enrollment and participation did not rebound to pre-pandemic numbers. Additionally, the implementation support staff member was not able to complete their work due to the pandemic. On-site support likely requires a longer lead time with anticipated larger and more sustained results compared to the other interventions; however, we were unable to evaluate this due to the COVID-19 shut-down. Pre-intervention data collection coincided with EMR transition at some sites, resulting in a limited ability to identify pre-intervention PrEP prescription numbers for some participants. Another limitation is the lack of a control group, and as such, the study design does not enable the ability to demonstrate causality given the potential for secular trends in PrEP-prescribing behavior. However, our findings are not thought to be due to natural increases in PrEP prescriptions over time given widespread disruption in PrEP services during the COVID-19 pandemic causing reductions in PrEP access during the time period in which this study’s intervention took place [33]. Lastly, we do not have long term retention and follow up data to understand whether the intervention had a lasting impact on PrEP-prescribing behavior, as well as whether patients continued on PrEP. These are important areas of future research.

In conclusion, a model designed to build from existing STI diagnosis and treatment processes in our health system’s primary care and women’s health clinics was successful in increasing the number of PrEP prescribers, PrEP prescriptions provided, and patients on PrEP within a large municipal public healthcare network in New York City.

## Figures and Tables

**Figure 1 viruses-15-01365-f001:**
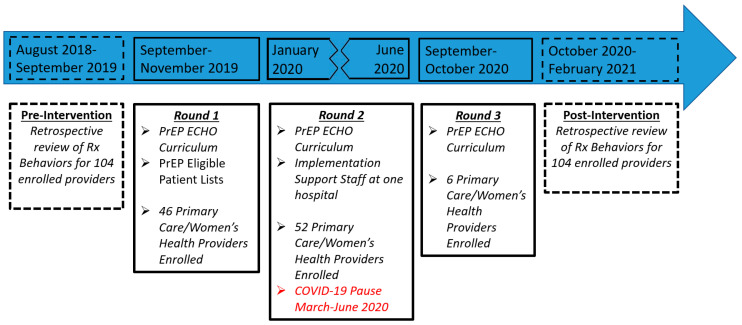
Intervention components and timeline.

**Table 1 viruses-15-01365-t001:** Number of PrEP prescriptions by providers, pre-intervention compared to post-intervention (n = 104).

	Pre-Intervention	Post-Intervention
Total number of prescriptions	32	297
Prescriptions by provider, mean (SD)	0.30 (1.09)	2.85 (5.17)
Number of individual PrEP patients	19	128

**Table 2 viruses-15-01365-t002:** Bivariate analysis of volume of prescribers and prescriptions, pre-intervention compared to post-intervention.

Department	Pre-Intervention, n (% or Mean)	Post-Intervention, n (% or Mean)	*p* Value
Women’s health prescribers (n = 24)	2 (8.3%)	10 (41.7%)	0.013
Women’s health, number of prescriptions	3/24 (0.125 Rx/provider)	43/24 (1.79 Rx/provider)	0.004
Primary care prescribers	10/80 (12.5%)	41/80 (51.3%)	<0.001
Primary care, number of prescriptions	29/80 (0.363 Rx/provider)	247/80 (3.09 Rx/provider)	0.009

**Table 3 viruses-15-01365-t003:** Number of providers with any PrEP prescriptions, pre-intervention compared to post-intervention.

Intervention Group or Type	Number of Participants	Participating Providers Prescribing PrEP Pre-Intervention n (%)	Participating Providers Prescribing PrEP Post-Intervention n (%)
Women’s Health	24	2 (8%)	10 (42%)
Primary Care	80	10 (13%)	41 (51%)
Completed ECHO	35	4 (11%)	18 (51%)
Curriculum Only *	69	8 (12%)	33 (48%)
PrEP Eligible List	30	6 (20%)	15 (50%)
Implementation Staff Support	12	0	8 (67%)

* <three sessions completed. PrEP-eligible list, implementation staff support, and completed ECHO are not mutually exclusive groups. Completed ECHO and Curriculum Only are mutually exclusive.

## Data Availability

Aside from the data presented within this article, no new data were created or analyzed in this study. Data sharing is not applicable to this article.

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
