# Peer review of "Successful Integration of HIV PrEP in Primary Care and Women’s Health Clinical Practice: A Model for Implementation"

_viruses, 2023, doi:10.3390/v15061365_

Round 1

Reviewer 1 Report

enjoyed reading this paper. 

Would have liked to know if all those who adopted PeP completed their course and recommended it to others

Author Response

Dear Reviewer,

Thank you for your review and comments.  Regarding your inquiry as to whether those who adopted PrEP continued on it and recommended it to others, we unfortunately do not have long-term follow up data on this intervention, however that is an area of active interested and planned future research.  We have added a sentence at the end of paragraph 5 on page 7 of the discussion where we mention limitations and lack of long-term retention and follow up data. Thank you again for your review.

Reviewer 2 Report

The manuscript: Successful integration of HIV PrEP in Primary Care and Women’s Health clinical practice; a model for implementation clearly shows that the prescription of PrEP has increased, even with relatively small sample size so I find this short communication suitable for publication.

There are no major issues detected, English language needs only minor editing

Author Response

Dear Reviewer,

Thank you for taking the time to review our manuscript and provide thoughtful feedback. We have reviewed and edited the manuscript as per your suggestion with attention to language and grammar.  Thank you again for your review. 

Reviewer 3 Report

Thank you for the opportunity to review “Successful integration of HIV PrEP in Primary Care and Women’s Health clinical practice: A model for implementation”. The paper reports an important advance in maximising the provision of PrEP in mainstream care settings. There are a number of typographic and layout problems which I will leave to the journal editors to resolve (although I will point out that the semicolon in the title should be a colon, the typo of “Februrary” in the abstract, and the conflation of names with degrees in the suggested citation block). Overall the paper is well set out, concise where it needs to be, and includes most of the relevant information, although I will suggest some additions below for readers not familiar with the current New York City HIV situation. Therefore, suggestions in this review should be considered by way of friendly amendments.

That said, in my opinion there are far too many tables and figures in this paper. Many of these provide detail that will be useful for the funders and implementers of the intervention, but only a few of them are useful as they stand to an external reader contemplating replicating such an intervention. I strongly encourage the authors to remove Figure 1 completely, as it provides no useful information that is not found elsewhere (and has no baseline data to provide comparison). Tables 1 and 2 are useful. Since the paper does not compare outcomes by intervention group or type, Tables 3 and 4 are not relevant to this paper, and are simply distracting. If Table 3 is included, it will require the addition of relevant text, and the headings in the table need to be consistently formatted. Just because a paper has lots of data or numbers does not make it useful , important, or more scientific. The authors’ single message—provide education to providers and you will improve the number of PrEP prescribing providers and the number of scripts—is powerful enough to stand alone. It would be useful to know if the ongoing mentoring made a difference, of course. Most of the other data I found unnecessary and distracting, if not confusing—for instance in Table 3, were the providers referred to in columns 3 and 4 subsets of the number of participants? If so, head these columns ‘Participant providers…’; of course this may create confusion with the bottom two rows. You see what I mean about these tables being confusing. If the purpose of the paper were to compare the efficacy of the intervention by group, then these tables/data would be useful. If that were the focus of the paper, however, then we would need to know why particular groups had different outcomes compared to other groups. But that is not the primary goal of the paper.

Now to more minor points. The latest date for PrEP data is 2019, over four years ago. Is there any later data? If not, then ‘the latest date for which data are available’ may be a useful insertion. I was interested that there are 14 000 NYers living with HIV, and 35 000 NYers who would benefit from prevention interventions; since this paper is focused on women’s health providers, it would be interesting and useful to tell us what proportion of these populations are women, and perhaps a little about the profile of the women at risk who are seeking HIV prevention interventions at these service delivery sites—are these mostly poor women? Are they routinely screened for HIV risk? Who initiated the risk conversation—the provider or the patient? The reader does not know the protocols of these clinics.

The authors report that the intervention was a telementoring education and support program, but we don’t know how the six biweekly 1 hour PrEP  sessions were delivered—did people gather around a screen in a room with small discussion groups? Was this entirely delivered online, with online discussion groups? Something else? I think delivery of the intervention is the kind of thing that readers of this paper will want more detail about.

If there were a clear research question stated at the beginning—something like ‘does an online intervention increase the number of PrEP providers and prescriptions?’—then the discussion section would have a focus and an opportunity to provide a clear answer to that urgent question. They would also avoid the inappropriate conclusion that integrating PrEP services into these clinics “is necessary”—that is a broad claim that is not supported by this paper: to do that would be to look at incident rates of new HIV infections on different patient populations. I think it would be much more supportable to say that where it is desirable to integrate PrEP services into primary care and women’s health clinics, providing a telemonitored intervention and mentoring to providers is an effective way of increasing the number of competent providers and the number of scripts written to women patients. I note that there is no detail about the cost (or cost-effectiveness) of this intervention, so it might be useful to say how much the intervention cost per provider education (or similar), so that planners who would like to adopt this model would have some idea of what they need to do to resource such an intervention. The authors do not say who pays for these PrEP meds in NYC, how many of these scripts were filled, or how adherence is monitored.

I appreciate the authors’ statement about causality and secular trends—this was something that occurred to me as I read the analysis. It is certainly possible that women may have become more demanding about HIV prevention as a result of lockdowns associated with Covid, or that providers might have become more PrEP-competent over time anyway as the effectiveness of PrEP become increasingly accepted.

Finally, the declaration about compliance with ethical standards is incomplete. The paper should not be published without this statement. There is no description of any ethical review in the paper itself, and that would be important to include. This is essentially a T1—T2 analysis and patients were not denied treatment they might otherwise have received, so the ethics of the study should be reasonably straightforward—but who reviewed it?  How safe is it for women patients to go home with a PrEP script in their pockets?

I encourage the authors to strengthen their paper with a view to how it might be read and received by other clinical providers who might wish to replicate this intervention. By stepping outside their own frame, they can tighten up the paper to make it cohesive and relevant to other providers.

Author Response

Dear Reviewer,

Thank you very much for taking the time to review our manuscript and provide thoughtful feedback.  Please see below a point-by-point response to your comments.

-Thank you for pointing out the typographical errors and we apologize for this.  We have reviewed and believe we have thoroughly edited the manuscript multiple times since and hope we have corrected all such errors. 

-We have taken your advice and removed the figure we had and created a new figure that outlines the various interventions and timeline of the intervention to hopefully provide additional clarity. 

-We kept Table 3 but added additional text as well as updated the column headings to specify 'Participating Providers'.  We have removed Table 4. 

-A statement related to the presented data being the latest available was added to the introduction. 

-We have taken your advice and included the % of female patients included in the HIV and PrEP eligible populations seen within our healthcare system. 

-As the safety net public hospital system in NYC the majority of our patients are uninsured or underinsured and have median household incomes below the federal poverty level. 

-We have added detail regarding the delivery of the curriculum.  It was up to individual sites as to how they wanted participants to watch- whether in a group or individually. Unfortunately, we do not have the granular details on how sites chose to deliver the curriculum at a site level.

-Thank you for the suggestion to clearly state the research question.  We have added this and reorganized the paper to support answering this question. We have also narrowed the claim stated by the paper to focus on the interventions we tested and the setting we used. 

-We added language explaining how PrEP is paid for in NYC (through insurance or Prescription assistance programs so that any patient interested/eligible can receive PrEP at no or little cost).  

-We have added detail to the ethics claim. 

Thank you again for your detailed and thoughtful feedback.  We hope that the additions/modifications we have made have adequately addressed your points and concerns.  

Reviewer 4 Report

The paper presents a useful intervention evaluation to increase prescription of PrEP to high risk individuals. The intervention was administered during a difficult time, given COVID-19 pandemic disruptions. The results are clear, the intervention improve PrEP prescription.

I had a difficult time comprehending the intervention components, what was delivered together, what was independent.  The results suggest that the intervention components were not administered across all physicians, but I can't see this detailed in the methods. More clarity must be offered in the methods overview to indicate if the intervention components can be considered as having impact independently or must only be considered as a comprehensive unit. Perhaps a layout diagram to show the times of different intervention component administration would help. If the components are independent, would the authors suggest that a less time intensive intervention, like a flag on the medical records of "eligible" individuals is able to increase prescriptions without the time intensive education component?

Also, there appears to be no control group. Can the authors speak to how the use of pre-post versus a control group limits internal validity? Historical events such as the development of new PrEP options, extensive advertising of PrEP, public health campaigns to inform consumers and providers, and possibly COVID-19 pandemic restrictions that may have  influenced consumer concerns about disease could all have influenced the outcomes.  

Author Response

Dear reviewer,

Thank you very much for taking the time to review our manuscript and provide thoughtful feedback.  

We have added additional detail and description in the methods and results regarding the different interventions as well as taken your suggestion and added a new figure (figure 1) to explain the various interventions over time.  

Your point on adding a PrEP eligible flag in the medical record is incredibly timely, as we have been working with our EHR IT department to develop exactly such a flag that has recently been put to use across our system. Our hopes- as you state- are that this makes it easier for providers to readily identify patients who could benefit from PrEP and initiate those discussions. 

We have added to the limitations regarding lack of control group and how this limits the interpretation of our results.  

Thank you again for your thorough and helpful review of our manuscript.